# Phenotypic Variability in Phelan–McDermid Syndrome and Its Putative Link to Environmental Factors

**DOI:** 10.3390/genes13030528

**Published:** 2022-03-17

**Authors:** Luigi Boccuto, Andrew Mitz, Ludovico Abenavoli, Sara M. Sarasua, William Bennett, Curtis Rogers, Barbara DuPont, Katy Phelan

**Affiliations:** 1Healthcare Genetics Program, School of Nursing, College of Behavioral, Social and Health Sciences, Clemson University, Clemson, SC 29634, USA; smsaras@clemson.edu; 2Laboratory of Neuropsychology, National Institute of Mental Health, National Institutes of Health, Bethesda, MD 20892, USA; mitza@mail.nih.gov; 3Department of Health Sciences, University Magna Graecia, 88100 Catanzaro, Italy; l.abenavoli@unicz.it; 4Division of Pediatric Gastroenterology, Hepatology, and Nutrition, Indiana University School of Medicine/Riley Hospital for Children, Indianapolis, IN 46202, USA; webjr@iupui.edu; 5Greenwood Genetic Center, Greenwood, SC 29646, USA; crogers@ggc.org (C.R.); dupont@ggc.org (B.D.); 6Genetics Laboratory, Florida Cancer Specialists &Research Institute, Fort Myers, FL 33916, USA; kphelan@flcancer.com

**Keywords:** Phelan–McDermid syndrome, phenotypic variability, *SHANK3*, 22q13 deletion, haploinsufficiency, *PNPLA3*, epigenetics, *BRD1*, pharmacogenomics, *CYP2D6*

## Abstract

Phelan–McDermid syndrome (PMS) is a multi-systemic disorder characterized by both genetic and phenotypic variability. Genetic abnormalities causing PMS span from pathogenic variants of the *SHANK3* gene to chromosomal rearrangements affecting the 22q13 region and leading to the loss of up to over nine megabases. The clinical presentation of individuals with PMS includes intellectual disability, neonatal hypotonia, delayed or absent speech, developmental delay, and minor dysmorphic facial features. Several other features may present with differences in age of onset and/or severity: seizures, autism, regression, sleep disorders, gastrointestinal problems, renal disorders, dysplastic toenails, and disrupted thermoregulation. Among the causes of this phenotypic variability, the size of the 22q13 deletion has effects that may be influenced by environmental factors interacting with haploinsufficiency or hemizygous variants of certain genes. Another mechanism linking environmental factors and phenotypic variability in PMS involves the loss of one copy of genes like *BRD1* or *CYP2D6*, located at 22q13 and involved in the regulation of genomic methylation or pharmacokinetics, which are also influenced by external agents, such as diet and drugs. Overall, several non-mutually exclusive genetic and epigenetic mechanisms interact with environmental factors and may contribute to the clinical variability observed in individuals with PMS. Characterization of such factors will help to better manage this disorder.

## 1. Introduction

Defining the contribution of environmental factors to the clinical presentation of a genetic disorder is often challenging. Many phenotypic features can be caused either by genetic alterations alone, environmental factors alone, or a combination of both. One example is the decreased growth rate in infancy, which can be due to failure to thrive (generally, a sign of genetic abnormalities), malnourishment (the result of a poor diet), or both. However, certain genetic disorders caused by large genomic alterations, with disruption or loss of several genes, leave little or no margin for a putative contribution of environmental factors. Theoretically, one of these disorders is Phelan–McDermid syndrome (PMS) (OMIM #606232): most cases are caused by chromosomal rearrangements leading to the loss of portions of the 22q13 region ranging from less than 100 kb to over 9 Mb [1,2,3,4]. In addition to chromosomal deletions of variable sizes, PMS can also be caused by pathogenic variants of the *SHANK3* gene, mapping in the telomeric region of 22q13.33 [5,6]. Such genetic heterogeneity is further complicated by cases with interstitial 22q13 deletions (not encompassing *SHANK3*) presenting with clinical features compatible with the PMS phenotype [7]. The clinical presentation of PMS is also variable; it can include moderate to profound developmental delay, neonatal hypotonia, delayed or absent speech, autistic traits, motor impairment, seizures, sleep disorders, gastrointestinal problems, renal disorders, dysplastic toenails, disrupted thermoregulation, and minor dysmorphic traits [1,2,3,4,5,6]. The difficulties of sorting the clinical profiles of individuals with interstitial 22q13 deletions from those with terminal deletions have led to a new classification of this disorder as PMS-*SHANK3* related, for cases with deletions or pathogenic variants affecting *SHANK3*, and PMS-*SHANK3* unrelated, for the remaining cases where *SHANK3* is preserved [8]. Although this classification promises to increase the diagnostic yield for this syndrome, it still does not explain the complexity of the genotype-phenotype correlation. Militating against easy genotype-phenotype assignment, some cases of PMS with large deletions present with relatively milder phenotypes and some cases with small deletions or pathogenic variants of *SHANK3* show severe features.

## 2. Possible Gene-Environment Interactions in Phelan–McDermid Syndrome

The widening of the investigational approach to 22q13 genes beyond the terminal region [9,10] and even to a genomic scale [11] has highlighted the potential contribution of various genes that can exert a modifying effect on PMS phenotypes. Moreover, the contribution of some of these genes can be, in turn, influenced by certain environmental factors.

In this commentary paper, we focus on a limited number of examples of the myriad of possible gene-environment interactions that can affect the PMS clinical presentation and we propose two major, but not necessarily mutually exclusive, mechanisms: (1) the loss of one copy of a 22q13 gene leads to haploinsufficiency or unmasks a pathogenic variant (hemizygosity), altering interactions with environmental factors in specific nodes of metabolic pathways; (2) the loss of a gene has a broad downstream effect, involving numerous genetic and/or epigenetic regulatory processes and affecting multiple pathways simultaneously.

### 2.1. Haploinsufficiency or Hemizygous Variants of 22q13 Genes

#### 2.1.1. *PNPLA3* and Liver Disease

The *PNPLA3* gene maps almost 7 Mb from the 22q telomere and can therefore be involved in both large terminal deletions [3,11] and interstitial deletions [7,9]. Consequently, its mechanism of gene-environment interaction may apply to both PMS-*SHANK3* related and PMS-*SHANK3* unrelated cases. The protein encoded by this gene—patatin-like phospholipase domain-containing protein 3 (PNPLA3), or adiponutrin—is a lipase that mediates triacylglycerol hydrolysis in adipocytes and hepatocytes; its expression is induced in the liver after feeding and during insulin resistance [12].

*PNPLA3* variants, particularly the p.Ile148Met change, have been associated with both alcoholic and non-alcoholic liver disease, with a higher predisposing effect in homozygous subjects than in heterozygous ones [12,13]. It has been reported that both homozygous and hemizygous *PNPLA3* variants can be detected in subjects with PMS, leading to steatohepatitis, hepatic fibrosis, and liver metabolic dysfunction, which in turn may cause a lack of response to pharmaceutical treatments [14].

The pathogenic effects of *PNPLA3* variants cause accumulation of lipids in the hepatocytes (macrovesicular steatosis) and may be amplified by environmental factors such as a high-calorie diet, alcohol, or oxidants (Table 1).

A similar mechanism can be proposed for the *PPARA* gene, mapping on 22q13.31 and encoding the peroxisome proliferator-activated receptor α: the complete deficiency of the Ppara protein in mouse hepatocytes leads to non-alcoholic fatty liver disease and liver inflammation in combination with a high-fat diet [15,16]. The abnormal lipid peroxidation induced by the lack of Ppara promotes the progression of liver injury to steatohepatitis and was proven to play a pivotal role in the adaptation to sepsis [16,17].

For this reason, individuals with PMS who have lost a copy of *PNPLA3* and/or *PPARA* should be screened for variants in the remaining copy and eventually follow a low-calorie, alcohol-free dietary regimen to reduce the risk for liver disease.

#### 2.1.2. *SCO2*, *TYMP*, *SULT4A1*, and Mitochondrial Dysfunction

Numerous genes involved in mitochondrial functions are located within the 22q13 region [18], and several studies have suggested that the haploinsufficiency of some of these genes, particularly *SCO2*, *TYMP*, and *SULT4A1* may contribute to the clinical presentation of PMS [7,9,10,11,19]. *SCO2* encodes a copper metallochaperone involved in the synthesis and maturation of the subunit II of cytochrome C oxidase, whose deficiency results in an increase in reactive oxygen species [10,18]. Pathogenic variants of this gene, such as the rare nonsense change p.Gln53* or the missense substitutions p.Ala259Val and p.Arg114His, can cause a severe form of autosomal dominant myopia (OMIM #608908) [20] or a fatal infantile cardioencephalomyopathy (OMIM #604377) [21]. *TYMP*’s product is a thymidine phosphorylase that promotes growth, angiogenic and chemotactic activities in endothelial cells. The reversible phosphorylase of thymidine catalysed by TYMP is also crucial for the rescue of pyrimidines for nucleotide synthesis. The lack of activity of this enzyme causes the autosomal recessive disorder mitochondrial DNA depletion syndrome 1 (MNGIE type) [10].

The neuron-specific sulfotransferase SULT4A1 localizes to the mitochondrial outer membrane and negatively regulates the interaction between SULT1A1 and SULT1A3, and through this mechanism, dopamine toxicity [22]. SULT4A1 protects against mitochondrial dysfunction induced by oxidative stress, presumably via the same mechanism [23]. Knockout of *Sult4a1* in mice causes progressive neurological degeneration and early death [24]. Loss of one copy of the *SULT4A1* gene has been implicated in PMS-*SHANK3* unrelated cases [7,9], but it may as well occur in PMS-*SHANK3* related cases carrying terminal deletions larger than 7 Mb [3,11].

Abnormal levels of reactive oxygen species and a decreased capacity of mitochondria to produce DNA and therefore adjust to different metabolic conditions can be further worsened by certain drugs, like chemotherapeutic agents or Nrf2-inhibitors, and lack of anti-oxidants introduced via diet and supplements [25,26] (Table 1).

#### 2.1.3. *SHANK3* and Inflammation

*SHANK3* maps in the subtelomeric region of chromosome 22q (22q13.33), encodes a multidomain scaffold protein of the post-synaptic density of several types of excitatory synapses, and its pathogenic alterations have been associated with several disorders, including, in addition to PMS-*SHANK3* related [5,6,8], autism spectrum disorder (ASD), schizophrenia, a Rett syndrome-like phenotype, and intellectual disability [27].

Disruption of protein-protein interactions at the postsynaptic density is a well-characterized model for the pathogenesis of ASD. SHANK proteins fit this model, as they are critical for the stability of glutamate receptors and cell adhesion proteins [28]. For this reason, several in vivo studies have targeted *Shank3* to investigate genetic and environmental factors that could affect synaptic stability and function, leading to ASD or other neurobehavioral disorders associated with this gene. One of such studies has shown that acute inflammation from a lipopolysaccharide challenge can acutely unmask behavioural deficits in *Shank3* haploinsufficient mice. The loss of social preference can be reversed by Trpv4 inhibition in the Nucleus Accumbens (NA) [29]. Early postnatal downregulation of NA Shank3 creates a similar behavioural deficit by shifting the membrane properties of dopamine type 1-expressing medium spiny neurons. The behavioural deficits are also rescued by Trpv4 inhibition, suggesting a mechanism in common between developmental and environmental influences.

Interestingly, the inflammatory response does not target directly *SHANK3* but rather exerts its influence via a gene-gene-environment model, where the environmental factor—the immune response—targets a gene, *TRPV4*, which maps on a different chromosome from *SHANK3* (12q24.11), and via its activation disrupts SHANK3 function at the post-synaptic density causing eventually reversible behavioural deficits. This model provides a valuable example of the intricacy of connections between genes and environmental factors and suggests that indirect influence on gene function may play an important role in the variability in genotype-phenotype correlation for PMS and other conditions.

### 2.2. Genome-Wide Effects

#### 2.2.1. *BRD1* and Epigenomic Dysregulation

Bromodomain-containing protein 1 (BRD1) is a component of the MOZ/MORF complex, which has a histone H3 acetyltransferase activity [10]. *BRD1* pathogenic variants have been associated with schizophrenia and functional studies on haploinsufficient animal models showed that this gene’s product modulates, via epigenetic regulation, behaviour, neurotransmission, and expression of genes involved in pathways associated with mental illness and molecular signaling processes [30]. Homozygous *Brd1* knockout mice have failure to thrive, seizures, decreased levels of histone H3 acetylation (H3K9ac, H3K14ac, and H3K18ac), and increased N-tail clipping [31].

Loss of one copy of *BRD1* alters genome-wide methylation, likely disrupting the epigenetic regulation of the expression of numerous genes (Table 1): the downstream effects are so broad that they significantly change genome-wide methylation and metabolic profiles in individuals with PMS [32]. For example, Thomford et al. [33] demonstrated how the methylation status of the promoter of another 22q13 gene previously discussed, *SCO2*, affects the penetrance of the effects of pathogenic variants in the non-deleted allele, suggesting a gene-epigene interaction leading to a potentially more severe phenotype. Epigenetic alterations have been associated with various conditions, including ASD [34], often present in individuals with PMS. Moreover, environmental factors such as medications have been shown to influence the risk for ASD and other neurodevelopmental conditions [35], plausibly via epigenetic modifications. It is plausible that epigenetic-mediated effects predisposing to ASD may play an additive or even synergistic role in PMS in combination with the disruption of *SHANK3*, the main 22q13 candidate gene for ASD and other neurological issues.

In conclusion, patients with 22q13 deletions encompassing *BRD1* may benefit from pharmacogenomic counseling to identify and eventually avoid drugs that can alter their epigenetic profile.

#### 2.2.2. *CYP2D6* and Pharmacogenomics

The *CYP2D6* gene encodes the cytochrome P450DB1 protein and maps in 22q13.2, therefore, just as discussed for *PNPAL3* and *SULT4A1*, the gene-environment interaction involving this gene can apply to large terminal deletions (PMS-*SHANK3* related cases) as well as interstitial deletions preserving the telomeric regions (PMS-*SHANK3* unrelated cases).

Cytochrome p450 proteins (CYP) constitute a superfamily of enzymes containing heme as a cofactor that function as monooxygenases and are involved in the metabolization of numerous types of drugs. For this reason, extensive research has been focused on the effect of variants in genes encoding CYP proteins on the pharmacokinetics of several compounds. Genes for CYP enzymes, such as *CYP2C9*, *CYP2C19*, *CYP2D6*, *CYP3A4*, and *CYP3A5,* have been classified as important pharmacogenes [36].

For *CYP2D6* in particular, established genotype-phenotype relationships allow for patients to be categorized into four groups based on the variant status: ultra-rapid metabolizers, extensive (normal) metabolizers, intermediate metabolizers, and poor metabolizers [37]. Among the classes of pharmaceutics whose efficacy is influenced by *CYP2D6* variants, there are compounds that can be used in the clinical management of individuals with PMS, like antidepressants, antipsychotics, and epileptics (Table 1). The potential co-occurrence of variants in other pharmacogenes may further affect the efficacy of these classes of drugs by increasing the disruption of the cytochrome p450 activity.

Therefore, physicians of subjects with PMS who carry deletions encompassing *CYP2D6* should consider genetic screening for variants in the preserved copy of this gene and other pharmacogenes, and eventually explore the possibility of pharmacogenomic counselling.

## 3. Critical Period Effects

Certain gene-environment interactions are limited to specific times of growth and development and thus not yet fully characterized. It is well understood, for example, that neurodevelopment undergoes critical periods and that these critical periods are temporal windows sensitive to disruptive influences from environmental factors [38]. Critical periods are associated with manifestations of neurodevelopmental disorders, affecting cognitive, social, and motor performance. Although environmental factors affecting critical periods may play an important role in PMS phenotype variability, the mechanistic details of this role remain largely unexplored. A step towards identifying potential gene-environment interactions is to identify genes of 22q13 that play a major, transient role in neurodevelopment, and that are likely sensitive to haploinsufficiency. An example is the *PHF21B* gene, whose sequence is highly constrained, indicating likely sensitivity to haploinsufficiency [9]. A change in isoform expression of this gene triggers the transition from neuronal progenitor cell to neuron during cortical neurogenesis through an epigenetic mechanism [39]. Neuronal differentiation has been associated with a wide variety of disruptive environmental factors [38], making *PHF21B* an obvious candidate for further investigation.

## 4. Conclusions

Although environmental factors play no causative role in the pathogenesis of PMS, there is evidence that they may affect the onset and/or severity of certain features. Characterization of their contribution to the syndrome’s phenotypes may improve clinical management and prognosis. This commentary paper provides some examples of gene-environment interactions that bear the potential of highlighting validated or hypothetical models for future investigation, suggesting molecules or pathways that could be used as biomarkers for early screening or prognosis, and identifying novel targets for precision treatments. The investigation of the impact of environmental factors on genetic disorders is still limited by the lack of information collected via functional studies: the complexity of the multiple interactions between different genes and various environmental factors is difficult to replicate in models designed to assess the impact of single variables. Future approaches will benefit from the data collected by omic studies, which will probably broaden our knowledge on the interactions between genes, proteins, and metabolites, identifying novel targets for environmental factors as well. Moreover, investigation of the gene-environment interactions in PMS provides valuable information on pathogenic mechanisms involving genes within and outside the 22q13 region and suggests an approach applicable to other multigenic disorders.

## Figures and Tables

**Table 1 genes-13-00528-t001:** Possible contributions of environmental factors to Phelan–McDermid syndrome clinical presentation.

**Gene (s)**	**Environmental Factor (s)**	**Mechanism**	**References**
*PNPLA3*, *PPARA*	Diet, drugs, inflammation	Hemizygous variants or complete loss of protein lead to liver disease, worsened by high-calorie foods, alcohol, drugs, and inflammation	[14,16,17]
*SCO2*, *TYMP, SULT4A1*	Diet, oxidants	Haploinsufficiency or hemizygous variants disrupt mitochondrial function, which can also be affected by certain nutrients/supplements and oxidants	[7,9,10,11,18,23]
*SHANK3*	Inflammation	Loss of a copy plus inflammation lead to hyperexcitability of medium spiny neurons	[28,29]
*BRD1*	Drugs	Loss of a copy alters epigenomic regulation, which is also influenced by certain drugs	[32,33,34,35]
*CYP2D6*	Drugs	Interactions with other components of cytochrome p450 affect responses to several drugs	[37]

## Data Availability

Not applicable.

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
