# Peer review of "Phenotypic Variability in Phelan–McDermid Syndrome and Its Putative Link to Environmental Factors"

_genes, 2022, doi:10.3390/genes13030528_

Round 1

Reviewer 1 Report

The manuscript entitled “Phenotypic variability in Phelan-McDermid syndrome and its putative link to environmental factors.” Submitted under the type “Commentary” is interesting but the heart of the topic is presented with insufficient support. Authors shall consider the following suggestions.

As the study is “Commentary”, It's worth noting that the authors should broaden the Discussion section to include a more personal point of view to consider. For example, in order to help readers grasp this complicated issue, they may respond to the following questions: what potential does this study have? What are the information gaps, and how are they being addressed by researchers? What do you think this concept will look like in the next five years? It would be incredibly intriguing for the readers, we believe.

How do the authors claim this statement “The PNPLA3 gene maps almost 7 Mb from the 22q telomere and can therefore be involved in both large terminal deletions and interstitial deletions.” Without appropriate supporting literature.

Authors shall incorporate specific Pathogenic variants of SCO2 gene that can cause a severe form of autosomal dominant myopia. This shall give precise idea about the  Pathogenic variants of SCO2 gene associated with autosomal dominant myopia.

Authors have to support with appropriate literature to support the presented statements like “but it may as well occur in PMS-SHANK3 related cases carrying terminal deletions larger than 7 Mb.”

Postsynaptic scaffolding proteins and their impact on ASD shall be discussed with support of literature like “Environmental and Genetic Factors in Autism Spectrum Disorders: Special Emphasis on Data from Arabian Studies” for best presentation in the section, SHANK3 and inflammation.

Table 1 is incomplete and lack of appropriate literature. A column with the appropriate reference shall be added for backup the genes and environmental factors presented in the “Table 1. Possible contributions of environmental factors to Phelan-McDermid syndrome clinical presentation.”  

Author Response

The manuscript entitled “Phenotypic variability in Phelan-McDermid syndrome and its putative link to environmental factors.” Submitted under the type “Commentary” is interesting but the heart of the topic is presented with insufficient support. Authors shall consider the following suggestions.

As the study is “Commentary”, It's worth noting that the authors should broaden the Discussion section to include a more personal point of view to consider. For example, in order to help readers grasp this complicated issue, they may respond to the following questions: what potential does this study have? What are the information gaps, and how are they being addressed by researchers? What do you think this concept will look like in the next five years? It would be incredibly intriguing for the readers, we believe.

  • We thank the Reviewer for the insightful comment, we added the following paragraph to the Conclusions to address these questions: “This commentary paper provides some examples of gene-environment interactions that bear the potential of highlighting validated or hypothetical models for future investigation, suggesting molecules or pathways that could be used as biomarkers for early screening or prognosis, and identifying novel targets for precision treatments. The investigation of the impact of environmental factors on genetic disorders is still limited by the lack of information collected via functional studies: the complexity of the multiple interactions between different genes and various environmental factors is difficult to replicate in models designed to assess the impact of single variables. Future approaches will benefit from the data collected by omic studies, which will probably broaden our knowledge on the interactions between genes, proteins, and metabolites, identifying novel targets for environmental factors as well.”

How do the authors claim this statement “The PNPLA3 gene maps almost 7 Mb from the 22q telomere and can therefore be involved in both large terminal deletions and interstitial deletions.” Without appropriate supporting literature.

  • We added references to support the statement.

Authors shall incorporate specific Pathogenic variants of SCO2 gene that can cause a severe form of autosomal dominant myopia. This shall give precise idea about the  Pathogenic variants of SCO2 gene associated with autosomal dominant myopia.

  • We added the specific pathogenic variants described in the cited article in the following sentence: “such as the rare nonsense change Gln53* or the missense substitutions p.Ala259Val and p.Arg114His”.

Authors have to support with appropriate literature to support the presented statements like “but it may as well occur in PMS-SHANK3 related cases carrying terminal deletions larger than 7 Mb.”

  • We added references to support the statement.

Postsynaptic scaffolding proteins and their impact on ASD shall be discussed with support of literature like “Environmental and Genetic Factors in Autism Spectrum Disorders: Special Emphasis on Data from Arabian Studies” for best presentation in the section, SHANK3 and inflammation.

  • We added a sentence to further discuss the role of protein-protein interactions at the synaptic levels in the pathogenesis of ASD and we cited “Environmental and Genetic Factors in Autism Spectrum Disorders: Special Emphasis on Data from Arabian Studies”.

Table 1 is incomplete and lack of appropriate literature. A column with the appropriate reference shall be added for backup the genes and environmental factors presented in the “Table 1. Possible contributions of environmental factors to Phelan-McDermid syndrome clinical presentation.” 

  • We added a column with the references.

Reviewer 2 Report

This manuscript is well appropriate as commentary.

As minor point: page 3, lines 122-125 requires references.

Author Response

This manuscript is well appropriate as commentary.

  • We deeply thank the reviewer for the positive comment.

As minor point: page 3, lines 122-125 requires references.

  • We added the following references: Cores et al., 2020; Kim et al., 2022.

Round 2

Reviewer 1 Report

Thank you for responding to the reviewer's concerns in such a detailed manner; the revised version can be accepted